# Elemental Selenium in the Synthesis of Selenaheterocycles

Alexander V. Martynov 

A. E. Favorsky Irkutsk Institute of Chemistry, Siberian Division of The Russian Academy of Sciences, 1 Favorsky Str., 664033 Irkutsk, Russia; martynov@irioch.irk.ru

**Abstract:** An overview of the known methods of introducing selenium under the action of elemental selenium into the structures of various saturated, unsaturated, and heteroaromatic selenacycles containing C–Se, N–Se, B–Se, Ge–Se and P–Se bonds is presented. These methods include metal, iodine, bromine or chlorine exchange for selenium and the direct cyclization of 1-(2-bromoaryl)benzimidazoles, polyunsaturated hydrocarbons, acetylenes, propargylic amines, 3-halogenaryl amides, aryl amides, diazo-compounds, 2-aminoacetophenone, and the annulation of ethynyl arenes. Three- and four-component reactions utilizing elemental selenium as one of the components and leading to selenium-containing heterocycles are presented as well.

**Keywords:** elemental selenium; metal–selenium exchange; cyclization; annulation; three-component reactions; four-component reactions

## 1. Introduction

The synthesis of organoselenium compounds, and especially selenium-containing heterocycles, continues to be a very active research area since the 1980s, when the results for the synthetic ebselen, an organoselenium compound 2-phenyl-1,2-benzoselenazol-3-one, revealed promising antioxidant properties of this heterocycle [1,2]. Nowadays, a variety of these compounds are known, which demonstrate antimicrobial, biocidal, anti-inflammatory, antioxidant and free radical scavenging activities [3–11]. Among them, a number of organoselenium compounds have been found suitable for the treatment of the most common ailments—cardiovascular, cancer, viral diseases and AIDS [2,12–17]. Their practical application in medicine for the treatment of tumors and cancers is a subject of current intense interest [18–24]. In material science, the utilization of selenium-containing heterocycles in developing organic conductors, semiconductors, electroconducting materials, paramagnetics and optoelectronics is another area of current interest [25–31].

The selenorganic heterocycles are usually prepared by the direct introduction of selenium into the organic scaffold or by exchange with another atom. Over the last half century, a number of selenylating agents have been introduced into the practice, including nucleophilic $H_2Se$, $NaHSe$, $Na_2Se$, $Li_2Se$, $KSeCN$, $(Me_3Si)_2Se$, electrophilic organylselenyl halides, selenium di- and tetrahalides [32–38]. But the most direct synthetic way to obtain selenaheterocycles consists of introducing elemental selenium into the parent organic molecule. Elemental selenium lacks the drawbacks of other selenylating agents such as toxicity, difficulty of preparation and handling as well as instability. At the same time, since this approach leads to the synthesis of selenium heterocycles by excluding additional manipulations with selenium such as, for instance, the generation of sodium or potassium selenides or diselenides $Na_2Se$, $K_2Se$, $Na_2Se_2$, $K_2Se_2$ or selenium di- or tetrahalides $SeX_2$, $SeX_4$ it seems quite prospective.

In 2019, a small review [39] was published in which some examples of elemental selenium introduction into the molecules of different heterocycles were presented. Quite a number of reviews depicting the syntheses of various selenium-containing compounds also included examples of the selenacyles formation due to the use of elemental selenium [40–51]. A fairly extensive overview of the use of elemental selenium for the syntheses of different

classes of organoselenium compounds, including various heterocycles, were presented by Ma et al. in 2021 [52] and Guo et al. in 2022 [53]. However, the use of elemental selenium for the synthesis of heterocycles has neither been considered in the literature as a separate subject nor in more detail. This review presents the currently known data related to the synthesis of selenium-containing heterocycles using elemental selenium as a selenylating reagent, which will allow us to establish the limits of the use of elemental selenium in the construction of various heterocyclic systems.

## 2. Synthesis of Selenium-Containing Heterocycles by Metal–Selenium Exchange in Cyclometallated Derivatives of Olefins, Allenes, Acetylenes and Aromatics

The exchange of metal for selenium in metallacyclopentanes **1**, -cyclopent-2-enes **2** and –cyclopenta-2,4-dienes **3**, generated in situ by Dzhemilev reaction (M = Al [54–57] and Mg [57–59]) through the catalytic cycloalumination or cyclomagnesation of alkenes and alkynes in the presence of catalytic amounts of Ti and Zr complexes, can be regarded as one of the earliest methods for the introduction of elemental selenium to heterocyclic compounds. Reactions result in a formation of selenium-containing five-membered saturated tetrahydroselenophenes **4**, and unsaturated dihydroselenophenes **5** and selenophenes **6** [54,56,59,60] (Scheme 1).

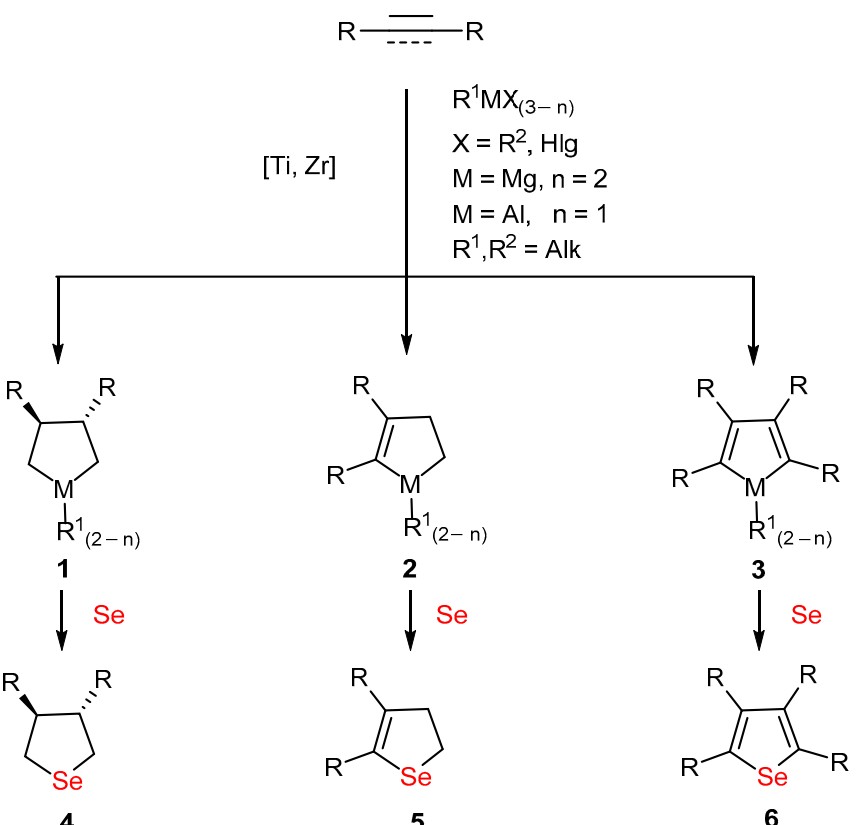

**Scheme 1.** Metal–selenium exchange in metallacyclopentanes **1**, -cyclopent-2-enes **2** and -cyclopenta-2,4-dienes **3**.

Based on this methodology, Dyakonov et al. [60] developed a one-pot synthesis of fused five-membered selenium heterocycles via the cyclometallation of methylenecyclobutane **7** and allenes **8a,b** using alkyl derivatives of Al and Mg. The reaction of the resulting alumina- and magnesacarbocycles with elemental selenium afforded various spiro-, bi- and tricyclotetrahydroselenophenes **9a,b,c** and bi- and tricycloselenophenes **10a,b** in high yields (Scheme 2).

**Scheme 2.** Cyclometallation of methylenecyclobutane **7** and allenes **8a,b**.

The cycloalumination of cyclotetradeca-1,8-diyne **11** in the presence of Zr catalyst (Cp$_2$ZrCl$_2$) involved both triple bonds of the diyne to present the isomeric tricyclic bisa-luminacyclopentenes **12** and **13** in a 1:1 ratio in a 91% yield. The reaction of the latter with an excess of elemental selenium in boiling benzene afforded a mixture of the regioisomeric 8,20-diselenatricyclo [15.3.01,17.07,11]eicosa-1(17),7(11)-diene **14** and 8,18-diselenatricyclo [15.3.01,17.07,11]eicosa-1(17),7(11)-diene **15** in a 1: 1 ratio and a 69% total yield [61] (Scheme 3).

**Scheme 3.** Cycloalumination of cyclotetradeca-1,8-diyne **11** with consequent Al–Se exchange.

The exchange of mercury for selenium in a mercury derivative of biphenyl **18** prepared by the treatment of diiodobiphenyl **16** with lithium and mercury chloride (II) at 200 °C led to dibenzoselenophene **6a** [62] (Scheme 4). Reaction proceeds through the intermediate formation of a dilithium derivative of biphenyl **17**.

**Scheme 4.** Hg–Se exchange in a mercury derivative of biphenyl **18**.

### 3. Synthesis of Selenium-Containing Heterocycles via Lithium–Selenium Exchange in Lithium Derivatives of Organic Compounds

The most developed method of introducing selenium into a heterocycle molecule is lithium–selenium exchange, which sometimes presents results that are difficult to achieve by other methods.

For instance, if to treat the intermediate dilithium biphenyl **17** mentioned above (Scheme 4) with elemental selenium in air, instead of mercury chloride, another product, dibenzo[1,2]diselenine **19,** is formed at the expense of direct lithium–selenium exchange, followed by aerial oxidation [62] (Scheme 5).

**Scheme 5.** Li–Se exchange in dilithium biphenyl **17**.

The reaction of the highly crowded trisilylmethyllithium compound $(PhMe_2Si)_3CLi$ with elemental Se resulted in a variety of products, which include the triselane $[(PhMe_2Si)_3CSe]_2Se$, the unexpected diselane $[(PhMe_2Si)_2HCSe]_2$ and the novel heterocycle *s*-tetraselenane $[(PhMe_2Si)_2CSeSe]_2$ **20** [63] (Scheme 6). The structure of the latter could be elucidated from the NMR spectroscopic data and was confirmed by the crystal structure, which displays the SeSeCSeSeC cycle in twist form.

**Scheme 6.** Formation of *s*-tetraselane **20** by Li–Se exchange in $(PhMe_2Si)_3CLi$.

The 1,2-diselenine-containing-fused π-conjugated compounds **24** were synthesized, starting from bis(*o*-haloaryl)diacetylenes **21** via a one-pot intramolecular triple cyclization reaction [64]. Further deselenation of 1,2-diselenine under the action of Cu afforded five-membered heteroacenes **25** [64] (Scheme 7).

**Scheme 7.** Triple cyclization of bis(*o*-haloaryl)diacetylenes **21** by action of BuLi/Se.

The formation of heterole-1,2-diselenin-heterole tricyclic structures **24** in this work was explained by the three-stage process, including the dilithiation of **21** with *t*-BuLi in THF, followed by trapping with elemental selenium to produce the dianionic species **22**. In the second step, the anionic centers attack the inner carbon atoms of the diacetylene moiety, generating a new dianionic species which is trapped with the remaining elemental selenium to afford the doubly cyclized dianionic intermediate **23**. The fused 1,2-diselenines **24** were obtained in the final step by the oxidation of the latter with potassium ferricyanide (III) in a 1 M NaOH aqueous solution (Scheme 7).

Similarly, the treatment of triphenylene derivative **26** with n-BuLi, followed by elemental selenium, afforded a 70% yield of heteroacene **27**, which was quantitatively transformed to the triselenasumanene derivative **28** by a solid-state deselenation over copper powder [65] (Scheme 8).

(1) TMEDA, n-BuLi (10 equiv.), 60 °C, 3 h; (2) Se (10 equiv.), −78 °C to r.t.; (3) Cu powder (10 equiv.), 200 °C, 2 h

**Scheme 8.** Synthesis of heteroacene **27** by action of BuLi/Se on triphenylene derivative **26**.

The dilithiation of 1,4-dibromo-2,5-bis(phenylethynyl)benzene **30a** with *t*-BuLi followed by treatment with selenium powder resulted in a synthesis of the corresponding 2,6-diphenybenzo [1,2-b:4,5-b']diselenophene **31a** (DPh-BDS) [66]. The parent **30a** was prepared in this work by the iodation of 1,4-dibromobenzene **29** and subsequent Sonogashira coupling with phenylacetylene [66] (Scheme 9). Other BDS derivatives **31b–d** with biphenyl-, p-hexylphenyl- and trimethylsilyl subsistents can also be synthesized by the same method using the corresponding acetylenes [66–70] (Scheme 9).

**Scheme 9.** Synthesis of BDSs **31** by action of *t*-BuLi/Se on 1,4-dibromo-2,5-bis(organylethynyl)benzenes **30**.

The treatment of disubstituted acetylene, 2,2′-dibromodiphenylacetylene **32a**, with *tert*-butyllithium followed by elemental selenium insertion in the Li derivative of acetylene **32a** resulted in a intramolecular ring closure to afford [1]benzoseleno [3,2-b][1]benzoselenophene **33a** [71] (Scheme 10). According to this procedure, [1]benzothieno [3,2-b][1]benzothiophene **33c** and [1]benzotelluro [3,2-b][1]benzotellurophene **33c** were also obtained.

**Scheme 10.** Annulation of 2,2′-dibromodiphenylacetylene **32a** under action of *t*-BuLi/Se as well as *t*-BuLi/Te, *t*-BuLi/S.

Dinaphtho [1,2-b:2′,1′-d]selenophene **35** was prepared by the exchange of lithium for selenium in the lithium derivative of sulfonamide generated by the treatment of sulfonamide **34** with n-BuLi in tetramethylethylenediamine (TMEDA). Furthermore, the diselenide **36** was formed in this reaction [72] (Scheme 11).

**Scheme 11.** Synthesis of dinaphtho [1,2-b:2′,1′-d]selenophene **35** by Li–Se exchange in lithium derivative of sulfonamide **34**.

An ebselen analog **39** was synthesized by treatment with selenium in THF of the lithium derivative of bisdihydrooxazole **37**, produced by the interaction of the latter with LDA in the presence of TMEDA [73,74]. It is assumed that the product **39** is formed due to the spontaneous disproportionation of the intermediate diselenide **38** (Scheme 12).

**Scheme 12.** Synthesis of ebselen analog **39** by Li–Se exchange in lithium derivative of bisdihydrooxazole **37**.

A general approach to ebselen and its derivatives **42** involving the use of elemental selenium was described. It includes the ortholithiation of benzanilides **40**, the subsequent insertion of elemental selenium into benzanilide-derived dianion **41** and the cyclization of selenium-containing dianion to ebselen derivatives **42** in yields up to 14% [75,76] (Scheme 13).

R = Ph, 4-MeC$_6$H$_4$, 3-MeC$_6$H$_4$, 2-MeC$_6$H$_4$, 4-MeC$_6$H$_4$CH$_2$, 2-MeC$_6$H$_4$CH$_2$

**Scheme 13.** Synthesis of ebselen and its derivatives **42** by Li–Se exchange in lithium derivatives of benzanilides **40**.

Selenazoloindoles **44** were prepared from the readily available *N*-alkynylindoles **43** via annulation through the introduction of n-BuLi and the sequential exchange of lithium for selenium under the action of elemental selenium. The simultaneous lithiation of triple bond to generate intermediate lithium selenolate results in the formation of the corresponding 3-alkylselanyl derivatives **44** on interaction with alkyl bromides [77] (Scheme 14).

$R^1$ = H, Me, OMe, CN; $R^2$ = Ph, 3-MeC$_6$H$_4$, 4-MeC$_6$H$_4$, 4-MeOC$_6$H$_4$, 4-ClC$_6$H$_4$, 3-F$_3$CC$_6$H$_4$, 4-FC$_6$H$_4$, 2-Naphthyl; $R^3$ = Pr, Bu, heptyl, Bn

**Scheme 14.** Synthesis of selenazoloindoles **44** by annulation of *N*-alkynylindoles **43** under action of BuLi/Se.

Treatment of 4,5,11,12-tetrabromo-*N*,*N*′-di-*n*-butyl-2,7,9,14-tetrakis(trimethylsilyl) tetraphenyleno [1,16-bcd:8,9-b′c′d′]dipyrrole **45** with excess of BuLi at −78 °C, and then with elemental selenium in THF at room temperature afforded diazadiseleno [8]circulene **46** thanks to the exchange of lithium for selenium in the lithium derivatives formed in the first stage [78] (Scheme 15).

**Scheme 15.** Synthesis of diazadiseleno [8]circulene **46** from 4,5,11,12-tetrabromo-*N*,*N*′-di-*n*-butyl-2,7,9,14-tetrakis(trimethylsilyl) tetraphenyleno [1,16-bcd:8,9-b′c′d′]dipyrrole **45** via Li–Se exchange.

Kobayashi et al. developed a synthetic approach to benzoselenazole-2(3H)-thiones **50**, 2-(alkylsulfanyl)benzoselenazoles **51** and S-(benzoselenazol-2-yl)thiocarboxylates **52** [79] (Scheme 16). Treatment of 1-bromo-2-isothiocyanatobenzene **47** with BuLi to produce 2-lithiophenyl isothiocyanates **48**, which were further reacted with selenium powder, yielded lithium benzo-1,3-selenazole-2-thiolate **49**. The quenching of this anion with the corresponding nucleophiles afforded the above benzoselenazoles **50**–**52**.

**Scheme 16.** Synthesis of lithium benzo-1,3-selenazole-2-thiolate **49** from 1-bromo-2-isothiocyanatobenzene **47** by action of BuLi/Se and consequent transformation of thiolate **49** into thiones **50**, sulfides **51** and thiocarboxylates **52**.

## 4. Synthesis of Selenium-Containing Heterocycles by Exchange of I or Br for Se

The exchange of halogens for selenium is another powerful method for the preparation of selenium-containing heterocycles.

Thus, diarylselenophenes **6** were obtained in 49–90% yields by the base-catalyzed Se–I exchange reaction of diaryliodonium salts **53** in DMSO at 80 °C [80] (Scheme 17). Diaryliodonium salts with both electron-rich and electron-deficient substituents can be used in this reaction.

$R^1$ = H, Me, Cl
$R^2$ = H, Me, t-Bu, Cl, Br, CF₃, CN, NHAc, Ph, 4-ClC₆H₄, 2-MeC₆H₄

**Scheme 17.** Synthesis of dibenzoselenophenes **6** by Se–I exchange in diaryliodonium salts **53**.

The annulation of ortho-alkenyl aryliodides **55** under the action of elemental selenium in the presence of CuI results in substituted benzoselenophenes **56** [81,82] (Scheme 18). The starting aryliodides **55** have been prepared here by the addition of arylzinc reagents **54** to alkynes in the presence of the cobalt–Xantphos complex to form *o*-alkenyl arylzinc intermediates and the subsequent substitution of the zinc substituent for iodine under the action of I₂ [82].

**Scheme 18.** Synthesis of benzoselenophenes **56** by annulation of ortho-alkenyl aryliodides **55** via Se–I exchange.

Similarly, the Cu-catalyzed reaction of 2-(2-iodophenyl)-1*H*-indoles **57** and Se powder in DMSO at 110 °C affords benzoselenopheno [3,2-b]indole derivatives **58** [83] (Scheme 19).

**Scheme 19.** Synthesis of benzoselenopheno [3,2-b]indoles **58** by annulation of 2-(2-iodophenyl)-1*H*-indoles **57** via Se–I exchange.

2-(2-Iodophenyl)imidazo [1,2-a]pyridine derivatives **59a** in similar conditions, under the action of elemental selenium, result in novel benzo[b]selenophene-fused imidazo [1,2-a]pyridines **60** [84] (Scheme 20). Both intramolecular cyclizations involve the Ullmann-type Se-arylation and C(sp$^2$)–H selenation reactions. However, reaction with imidazopyridine derivatives **59a** in contrast to reaction with indoles **57** proceeds under aerobic conditions. Products were prepared here in moderate-to-high yields.

**Scheme 20.** Synthesis of benzo[b]selenophene-fused imidazo [1,2-*a*]pyridines **60** by annulation of 2-(2-iodophenyl)imidazo [1,2-a]pyridine derivatives **59a** via Se–I exchange.

An alternative method for the synthesis of benzo[*b*]selenophene-fused imidazo [1,2-a]pyridines **60** through ligand- and base-free CuI-catalyzed cyclization of 2-(2-bromophenyl) imidazo [1,2-a]pyridine derivatives **59b** under the action of elemental selenium in air was also described [85] (Scheme 21).

A copper-catalyzed reaction between 2-bromobenzothioamides 2-Br-RC$_6$H$_3$C(S)NHR$^1$ (R = H, 5-Me, 5-Cl, 3-Me, etc.; R$^1$ = Ph, pyridin-2-yl, 9H-fluoren-2-yl, etc.) **61** and Se involves sulfur rearrangement and enables access to benzothiaselenoles **62** in the presence of Cs$_2$CO$_3$. In the absence of Se, the reaction affords dibenzodithiocines **63** (R = H, 3-OMe, 2-Me, etc.) via two consecutive C(sp$^2$)-S Ullmann couplings [86] (Scheme 22).

**Scheme 21.** Synthesis of benzo[b]selenophene-fused imidazo [1,2-a]pyridines **60** by annulation of 2-(2-bromophenyl) imidazo [1,2-a]pyridine derivatives **59b** via Se–Br exchange.

**Scheme 22.** Synthesis of benzothiaselenoles **62** by Se–Br exchange in 2-bromobenzothioamides **61** and consequent sulfur rearrangement.

A novel and efficient procedure for one-pot regio- and stereospecific synthesis of benzo [1,4,2]thiaselenazine 1,1-dioxides **66** via [Cu]-catalyzed ring closure reaction between *N*-alkynyl-2-iodobenzene sulfonamides **64** and elemental Se in *N*-methyl-2-pyrrolidone (NMP) at 90 °C for 20 h has been developed [87]. Its generality was illustrated by extension to the synthesis of seven-membered benzothiaselenazepines **67** from *N*-(3-phenylprop-2-yn-1-yl)-2-iodobenzene sulfonamides **65** (Scheme 23). The involvement of water in the reaction is demonstrated by the incorporation of 2D at the olefinic site by using $D_2O$ in place of water.

**64, 66**: R = Me, *t*-Bu, Ph; R$^1$ = Me; R$^2$ = n-Hex, Ph, 4-MeOC$_6$H$_4$, 4-MeC$_6$H$_4$, 3-FC$_6$H$_4$, 4-biphenyl

**65, 67**: R = Me, *t*-Bu, Ph; R$^1$ = Me, Ph

**Scheme 23.** Synthesis of benzo [1,2,4]thiaselenazine 1,1-dioxides **66** and benzothiaselenazepines **67** by Se–I exchange with ring closure in N-alkynyl-2-iodobenzene sulfonamides **64** and N-(3-phenylprop-2-yn-1-yl)-2-iodobenzene sulfonamides **65**.

A similar procedure was used for the synthesis of 2,3-dihydro-1,4-benzoxaselenines **69** from 2-iodoaryl propargyl ethers **68** [88] (Scheme 24).

R = C$_6$H$_3$, MeOC(O); R$^1$ = Ph, o-tolyl, 3-thienyl, 1-Np; R$^2$ = H, Me

**Scheme 24.** Synthesis of 2,3-dihydro-1,4-benzoxaselenines **69** by Se–I exchange and ring closure in 2-iodoaryl propargyl ethers **68**.

A CuBr$_2$-catalyzed annulation of 2-bromo-*N*-arylbenzimidamide **70** with selenium powder was shown to be a general convenient method for the preparation of benzo[d]isosel enazoles **71** in good yields [89] (Scheme 25). This synthetic strategy demonstrates good functional group tolerance. Furthermore, the corresponding products could be converted into *N*-aryl indoles **72** in the reactions with diarylacetylenes **32c** via rhodium$^{III}$-catalyzed *ortho*-C–H activation of the *N*-phenyl ring, providing an efficient approach for axial aromatic molecules.

**Scheme 25.** Synthesis of benzo[d]isoselenazoles **71** by CuBr$_2$-catalyzed annulation of 2-bromo-*N*-arylbenzimidamide **70** with selenium and conversion of **71** into *N*-aryl indoles **72**.

## 5. Synthesis of Benzoselenazoles by Cyclization of 1-(2-Bromoaryl)benzimidazoles under Action of Selenium

The ring-closure reaction of 1-(2-bromoaryl)benzimidazoles **73** with Se powder was promoted by Cs$_2$CO$_3$ in DMF at 150 °C and afforded novel tetracyclic heterocycles— benzimidazo [2,1-b]benzoselenoazoles **74** [90] (Scheme 26). As compared to the above methodology, the proposed mechanism of this reaction involves the deprotonation of the imidazole ring at the 2-position and C(Het)–Se bond formation. Consequent ring closure via the S$_N$Ar reaction by attack of the selenide anion on the phenyl group containing bromine generates the target tetracyclic molecule (Scheme 26). Single-crystal X-ray analysis of the parent benzimidazo [2,1-b]benzoselenoazole **74a** (R = R$^1$ = H) revealed that the tetracyclic ring is almost planar.

**Scheme 26.** Ring closure in 1-(2-bromoaryl)benzimidazoles **73** with Se powder.

## 6. Cyclization of Polyunsaturated Hydrocarbons under the Action of Elemental Selenium

Another method that can be regarded as one of the earliest for the synthesis of selenium-containing heterocycles is the cyclization of polyunsaturated hydrocarbons under the action of elemental selenium.

The selenation of tetraarylbutatrienes $R_2C=C=C=CR_2$ **75** (R = 4-$R^1C_6H_4$; $R^1$ = Me, H, Cl) in DMF in the presence of DBU afforded 1,2,5-triselenepanes **76**, while sulfurization resulted in 1,2,3,4,5-pentathiepanes **77**. The further degradation of 1,2,5-triselenepanes **76** resulted in the formation of benzoselenophene derivatives **78** [91] (Scheme 27).

**Scheme 27.** Selenation of tetraarylbutatrienes **75** with elemental Se to afford 1,2,5-triselenepanes **76** as compared to sulfurization.

[4+1]-Cycloaddition of elemental selenium to trifluoromethyl derivatives of 1,3-diene **79** in an autoclave without a solvent and in the presence of anhydrous trifluoroacetic acid anhydride (TFAA) as catalyst presented 2,4-substituted 2,5-dihydroselenophenes **80** [92] (Scheme 28).

**Scheme 28.** Synthesis of 2,5-dihydroselenophenes **80** by cyclization of 1,3-diene **79** under action of elemental Se.

The reaction of diaryldiynes **81** with elemental selenium, in contrast to 1,3-dienes, afforded—on heating to 170–230 °C—various diselenolodiselenole derivatives **82** [93] (Scheme 29).

Ar = thienyl, 4-C$_6$H$_{13}$OC$_6$H$_4$, 4-FC$_6$H$_4$, 4-MeC$_6$H$_4$COO

**Scheme 29.** Synthesis of diselenolodiselenole derivatives **82** by action of elemental Se on diaryldiynes **81**.

The direction of the reaction with diynes changes on the addition of hydrazine monohydrate and KOH to elemental selenium. Thus, the treatment of diphenyl diacetylene **81a** with Se/N$_2$H$_4$·H$_2$O/KOH system afforded 2,5-diphenylselenophene **6b** due to the generation of K$_2$Se in the reaction mixture [94]. Similarly, 1,3-butadiyne-bridged carbazole dimer **83** afforded selenophene-bridged carbazole dimer—isophlorin **84,** which, upon oxidation with MnO$_2$ in CH$_2$Cl$_2$, led to selenaporphyrin **85** [95] (Scheme 30).

**Scheme 30.** Formation of selenophene moieties in compounds **6b** and **84** by action of Se/N$_2$H$_4$·H$_2$O/KOH on diynes **81a** and **83**.

## 7. Selenophenes via Cyclization of Acetylenes under Action of Elemental Selenium

One of the best known methods for producing selenophenes is the interaction of terminal and disubstituted acetylenes with elemental selenium in benzene at elevated pressure. A reaction of 3-butyn-2-one **32b** with Se at 205–215 °C in $C_6H_6$ in stainless autoclave resulted in 2,4- and 2,5-diacetylselenophenes **6c** and **6d**, while tetraphenylselenophene **6e** was prepared in a similar method to that of PhC≡CPh **32c** [96] (Scheme 31). The proposed mechanism for the formation of selenophenes involves the intermediate generation of diselenins **86**, which, under reaction conditions, are subjected to deselenation.

**Scheme 31.** Formation of selenophenes **6c**, **6d** and **6e** by action of elemental Se on acetylenes **32b** and **32c**.

## 8. A Carbonylative Cyclization of Propargylic Amines with Elemental Selenium

Various 1,3-selenazolidin-2-ones **88** were prepared via a carbonylative cyclization of propargylic amines **87** with elemental selenium [97] (Scheme 32). In this process, as a safe and convenient solid CO source, benzene-1,3,5-triyl triformate (TFBen) was employed using *t*-BuOK as the promoter. A broad class of substrates was effectively transformed into the desired products under mild conditions.

**Scheme 32.** Synthesis of 1,3-selenazolidin-2-ones **88** by cyclization of propargylic amines **87** with elemental Se in the presence of benzene-1,3,5-triyl triformate (TFBen).

When in the same process with TFBen as a CO source where DBU was used as a reaction promoter while $n$-$C_4F_9I$ was used as the iodide source, a carbonylative cyclization

of propargylic amines **87** with elemental selenium afforded (*E*)-5-(iodomethylene)-1,3-selenazolidin-2-ones **89** in up to 95% yields [98] (Scheme 33).

**Scheme 33.** Synthesis of (*E*)-5-(iodomethylene)-1,3-selenazolidin-2-ones **89**.

An alternative method for the preparation of 5-alkylidene-1,3-selenazolin-2-ones **88** was described by Fujiwara et al. [99], who suggested the 5-*exo-dig* cyclization of selenolate intermediate **90** generated by the reaction of propargylic amines **87** with elemental selenium and CO using DBU as a promoter (Scheme 34).

R = H, Me, *i*-Pr, *n*-Bu, *t*-Bu, cyclohexyl, Ph; R$^1$ = H, Et, TMS, Ph

**Scheme 34.** Alternative method for synthesis of 1,3-selenazolin-2-ones **88** by cyclization of propargylic amines **87** with elemental Se and CO.

A similar CuI-catalyzed cyclocarbonylation of homopropargylamine **91** under the action of CO in the presence of elemental Se afforded the corresponding selenazinan-2-one derivative **92** [99] (Scheme 35).

**Scheme 35.** Synthesis of selenazinan-2-one **92** by cyclization of homopropargylamine **91** under action of elemental Se/CO.

## 9. Synthesis of 1,2,3,5,6,7-Hexaselenacyclooctane via Se–Cl Exchange in 1-Chloro-2,2-bis(diethylamino)ethene under Action of Elemental Selenium

The possibility of Se–Cl exchange in chloroethenes was demonstrated by the synthesis of 4,8-bis[bis(diethylamino)methylene]-1,2,3,5,6,7-hexaselenacyclooctane **94** [100,101]. Treatment of 1-chloro-2,2-bis(diethylamino)ethene **93** with elemental Se in refluxing benzene resulted in compound **94** in 60% yield (Scheme 36). Its structure was determined by XRD analysis. The compound **94** was shown to behave as 2,2-bis(diethylamino)-2-ethylium-1-diselenocarboxylate **95** toward a range of reagents. Thus, with di(methyl) acetylenedicarboxylate **32c**, it reacted to provide 1,3-diselenole **96** in high yield (Scheme 36). Evidence for the dissociation of **94** into **95** in solutions was provided by IR, UV/visible and $^1$H-, $^{13}$C- and $^{77}$Se-NMR spectra.

**Scheme 36.** Synthesis of 1,2,3,5,6,7-hexaselenacyclooctane **94** by Se–Cl exchange in 1-chloro-2,2-bis(diethylamino)ethene **93** under action of elemental Se.

## 10. Synthesis of Ebselen via Cyclization of 3-Halogenaryl Amides and Aryl Amides under Action of Selenium

A general method was developed for the synthesis of the aforementioned biologically important ebselen and related analogs **42** containing a Se–N bond. It involves an efficient copper-catalyzed selenium–nitrogen coupling reaction between various 2-chloro, 2-bromo, 2-iodo-arylamides **97** and selenium powder [102,103] (Scheme 37). This copper-catalyzed reaction tolerates functional groups such as amides, hydroxyls, ethers, nitro, fluorides and chlorides. The best results have been obtained by using a combination of potassium carbonate as a base, or iodo-/bromo-arylamide substrates and copper iodide catalyst.

$R^1$ = H, 5-$CH_3O$, 3-$CH_3O$, 3,4-$(CH_3O)_2$, 5-$CH_3S$, pyridyl
R = $C_6H_5$, $CH_2C_6H_5$, *n*-Bu, 2-$FC_6H_4$, 2-$ClC_6H_4$, 2-$BrC_6H_4$, 2-$CH_3OC_6H_4$, cyclohexyl

**Scheme 37.** Synthesis of ebselen and its analogs **42** by cyclization of 2-halogen arylamides **97** under action of elemental Se.

In order to prepare the ebselen analogs bearing an 8-quinolyl moiety, the arylamides **97** without halogen substituents at aryl moieties were also used. This efficient Ni-catalyzed selenation reaction was carried out in DMF at 120 °C in air and afforded for 24 h the corresponding benzoselenazole derivatives **42** in good yields [104] (Scheme 38).

$R^1$ = H, Me, Cl, Br, CN, $CO_2Me$, MeO, $CF_3$
R = quinolin-8-yl, 5-methoxyquinolin-8-yl

**Scheme 38.** Synthesis of ebselen and its analogs **42** by Ni-catalyzed cyclization of arylamides **97** with elemental Se.

## 11. Annulation of Ethynyl Arenes under Action of Selenium

Action of elemental Se on ortho-monoalkynyl-substituted perylene diimide (PDI) **98** in dimethylacetamide (DMA) at 140 °C resulted in highly regioselective heteroannulation to form selenophene-fused polycyclic product **99** [105] (Scheme 39).

**Scheme 39.** Annulation of ortho-monoalkynyl-substituted perylene diimide (PDI) **98** with elemental Se.

## 12. Formation of Heterocycles via Cyclization of Diazo-Compounds under Action of Selenium

The reaction of bis(diazo)octamethyldecane **100** with elemental selenium in DBU at 130 °C yielded 1,2-di-*tert*-butyl-3,3,6,6-tetramethylcyclohexene **101** as the major product along with *trans*-3,8-di-*tert*-butyl-4,4,7,7-tetramethyl-1,2-diselenocane **102**, while the analogous reaction of the reagent **100** with elemental sulfur in DBU resulted in *trans*-3,8-di-*tert*-butyl-4,4,7,7-tetramethyl-1,2-dithiocane **103** as the only product [106] (Scheme 40). The reaction of 3,9-bis(diazo)-2,2,4,4,8,8,10,10-octamethylundecane **104** with elemental selenium in DBU at 80 °C resulted in the formation of cyclic triselenide, *cis*-4,10-di-*tert*-butyl-5,5,9,9-tetramethyl-1,2,3-triselenecane **105** as the only identifiable product [106] (Scheme 40). The structures of the heterocycles **103** and **105** were confirmed by X-ray crystallography.

**Scheme 40.** Cyclization of bis(diazo)octamethyldecane **100** and bis(diazo)octamethylundecane **104** under action of elemental Se.

## 13. Formation of Selenazolines via Action of Selenium on *N*-Acyl-2-oxazolidinones

An efficient method for the preparation of chiral selenazolines (4,5-dihydro-1,3-selenazoles) **107** from *N*-acyl-2-oxazolidinones **106** and elemental Se in the presence of amine and hydrochlorosilane was suggested by Shibahara et al. [107] (Scheme 41). Suggested reaction mechanisms include the selenative rearrangement of *N*-acyl-2-oxazolidinones **106** and the elimination of O=C=Se species. A similar selenative rearrangement was observed in the reaction of free oxazolidinone **108** carried out under the same selenation condition and affording selenazolidinone **109** in moderate yield [107] (Scheme 42).

**Scheme 41.** Synthesis of chiral selenazolines **107** under action of elemental Se on *N*-acyl-2-oxazolidinones **106**.

**Scheme 42.** Synthesis of selenazolidinone **109** under action of elemental Se on oxazolidinone **108**.

## 14. Carbonylative Cyclization of 2-Aminoacetophenone under Action of Se and CO

2H-3,1-Benzoselenazin-2-one **111** was prepared in 58% yield by carbonylation of *o*-aminoacetophenone **110** with Se and CO (under 30 atm pressure) in the presence of N-methylpyrrolidone at 100 °C for 20 h [108] (Scheme 43). The reducing agent, $H_2Se$, was presumably formed by the reaction of selenium, carbon monoxide and water.

**Scheme 43.** Synthesis of 2H-3,1-benzoselenazin-2-one **111** by carbonylation of *o*-aminoacetophenone **110** with elemental Se/CO.

## 15. Synthesis of 1,3-Oxaselenoles from Carbonyl-Stabilized Sulfonium Ylides under Action of Selenium

Carbonyl-stabilized sulfonium ylides **112a,b** readily react with elemental selenium to afford 1,3-oxaselenole derivatives **113a,b** in good yields, thus providing a simple method for constructing these ring systems, which use easily accessible compounds as starting materials [109] (Scheme 44).

Scheme 44. Cyclization of sulfonium ylides **112a,b** under action of elemental Se.

## 16. Synthesis of 1,3-Diselenole-2-selones by Interaction of Terminal Acetylenes with Se, CSe$_2$ and BuLi

4-Methylthio-5-(2-methoxycarbonylethylthio)-1,3-diselenole-2-selone **116a** have been prepared in high yields from methylsulfanyl acetylene **32e** or 1-methylsylfanyl-1,2-dichloroe thylene **114** by their lithyation with 1 or 2 equivalents of BuLi to result in lithium acetylenide, followed by consequent reaction with elemental selenium and carbon diselenide, and finally with methyl 3-thiocyanateproponate **115** [110] (Scheme 45).

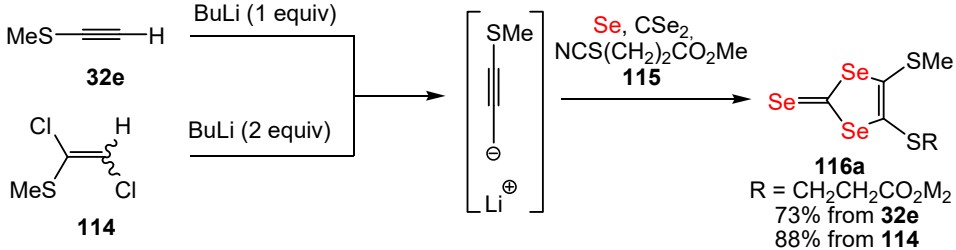

Scheme 45. Synthesis of 4-methylthio-5-(2-methoxycarbonylethylthio)-1,3-diselenole-2-selone **116a** by action of elemental Se/CSe$_2$ on lithum acetylenide generated from methylsulfanyl acetylene **32e** or 1-methylsylfanyl-1,2-dichloroethylene **114**.

4,5-Alkylenedichalcogeno-substituted 1,3-diselenole-2-selones **116** have been prepared in a similar way by a one-pot synthetic method, including the successive treatment of trimethylsilylacetylene **32f** with BuLi, Se, CSe$_2$ and finally, α,ω-bis(chalcogenocyanato)alkanes NCZ(CH$_2$)$_n$ZCN (Z = S, Se; n = 1–3) **117** [111] (Scheme 46).

Scheme 46. Synthesis of 4,5-alkylenedichalcogeno-substituted 1,3-diselenole-2-selones **116** from trimethylsilylacetylene **32f** under action of BuLi/Se/CSe$_2$.

## 17. Introduction of Selenium into Organic Molecule via Carbanion

The treatment of the mesyloxymethyl-substituted β-lactams **118a–c** with elemental selenium and *t*-BuOK in THF/DMF led to the synthesis of the *cis*-configurated biologically active isodethiaselenapenam **119** as well as isodethiaselenacephems **120a,b** [112]

(Scheme 47). The key step of this synthetic approach involved the addition of Se to the corresponding carbanions, followed by internal alkylation.

**Scheme 47.** Cyclization of mesyloxymethyl-substituted β-lactams **118a-c** with elemental Se/t-BuOK.

## 18. Formation of Heterocycles with Se–El Bonds (El = B, Ge, P)

### 18.1. Formation of Heterocycles with Se–B Bonds

The treatment of annulated 1,4,2,5-diazadiborinine **121** with elemental selenium resulted in the oxidative addition of selenium, which proceeded regioselectively at the boron centers of diborine **121** to present a bicyclo [2.2.2] molecule **122** with a B–Se–Se–B unit, which can be deemed a heavier analog of diboraperoxide [113] (Scheme 48).

**Scheme 48.** Formation of bicyclo [2.2.2] system **122** by treatment of 1,4,2,5-diazadiborinine **121** with elemental Se.

The reaction of diborene **123** with elemental selenium is shown to afford diboraselenirane **124** [114] (Scheme 49). This reaction is reminiscent of the sequestration of subvalent oxygen and nitrogen in the formation of oxiranes and aziridines; however, such reactivity is not known between alkenes and the heavy chalcogens. Although carbon is too electronegative to affect the reduction in elements with a lower relative electronegativity, the highly reducing nature of the B–B double bond enables reactions with $Se^0$.

**Scheme 49.** Synthesis of diboraselenirane **124** by action of elemental Se on diborene **123**.



The reductive insertion of elemental selenium into the B–B triple bond of the first stable diboryne **125** [115] under ultrasonic agitation has led to the synthesis of an Se-bridged cyclic compound containing boron stabilized by *N*-heterocyclic carbene (NHC) **126**. The three pairs of bonding electrons between the boron atoms in the triply bonded diboryne enabled a six-electron reduction reaction, resulting in a [2.2.1]-bicyclic system wherein bridgehead B atoms are spanned by three selenium bridges [116] (Scheme 50). Unfortunately, no yields have been reported for both compounds **124** and **126**.

**Scheme 50.** Insertion of elemental Se into B≡B bond of diboryne **125**.

Reaction of silaborene, $R_2Si=B(tmp)$ (R = SiMeBu-$t_2$, tmp = 2,2,6,6-tetramethylpiperidine) **127** with elemental selenium in THF afforded the novel three-membered ring product, selenasilaborirane **128**. In contrast, the oxidation of **127** under an $O_2$ atmosphere produced the four-membered ring, 1,3,2,4-dioxasilaboretane **129** [117] (Scheme 51). Compounds **128** and **129** were studied using XRD analysis.

**Scheme 51.** Synthesis of selenasilaborirane **128** by insertion of elemental Se into Si=B bond of silaborene **127** as compared to oxidation leading to 1,3,2,4-dioxasilaboretane **129**.

The treatment of the stable 1-phospha-2-boraacenaphthene **130** with elemental selenium afforded the unique heterocycle, 2-selena-1-phospha-3-boraphenalene **131** through the insertion of the selenium atom into a P–B bond of acenaphtene **130**. Further selenation of phenalene **131** led to 2-selena-1-phospha-3-boraphenalene-1-selenide **132** [118]. The unique dynamic behavior of phosphine selenide **132** in solution was explained by facile selenium exchange in the molecule (Scheme 52).

The carborane-fused heterocycles **134a–c** were prepared in good isolated yield via the reaction of carborane-fused zirconacyclopentane **133** with elemental selenium as well as with sulfur and tellurium [119] (Scheme 53). This approach represents a promising route to obtain functionalized carboranes that are difficult to access through conventional methods.

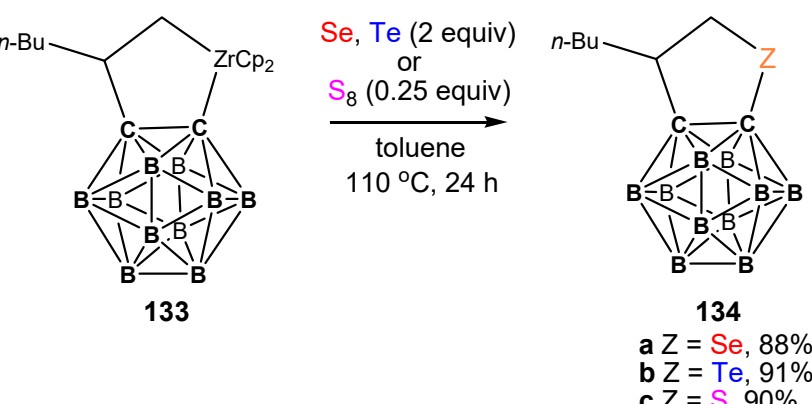

**Scheme 52.** Insertion of elemental Se into P–B bond of 1-phospha-2-boraacenaphthene **130**.

**Scheme 53.** Zr–Se(S,Te) exchange in carborane-fused zirconacyclopentane **133**.

### 18.2. Formation of Heterocycles with Se–Ge Bonds

The reaction of stable hafnocene-based bicyclo [2.1.1]hexene germylene **135** with elemental selenium provides access to 1,3-diselena-2,4-digermetane **137a** formed as a 87:13 mixture of *cis*- and *trans*-isomers (Scheme 54). This strongly colored four-membered germanium heterocycle, which is the formal dimer of a heavy ketone **136**, was characterized via NMR and UV spectroscopy as well as the results of an XRD analysis [120]. The same reactions were realized in the case of elemental sulfur and tellurium [120] (Scheme 54).

Alternatively, 2,2,4,4-tetrakis [2-(dimethylamino)phenyl]-1,3-diselena-2,4-digermetane **137d** was prepared by the reaction of {tris [2-(dimethylamino)phenyl]germyl}lithium (R₃GeLi) **138** with elemental selenium [121] (Scheme 55). The crystal structure of this heterocyclic compound has been determined by XRD analysis. The authors explained the formation of 1,3-diselena-2,4-digermetane either by an intermediate generation of germanselone and its intramolecular formal head-to-tail [2+2] cycloaddition or an intermolecular nucleophilic attack of the selenide ion at the germanium atom of another molecule.

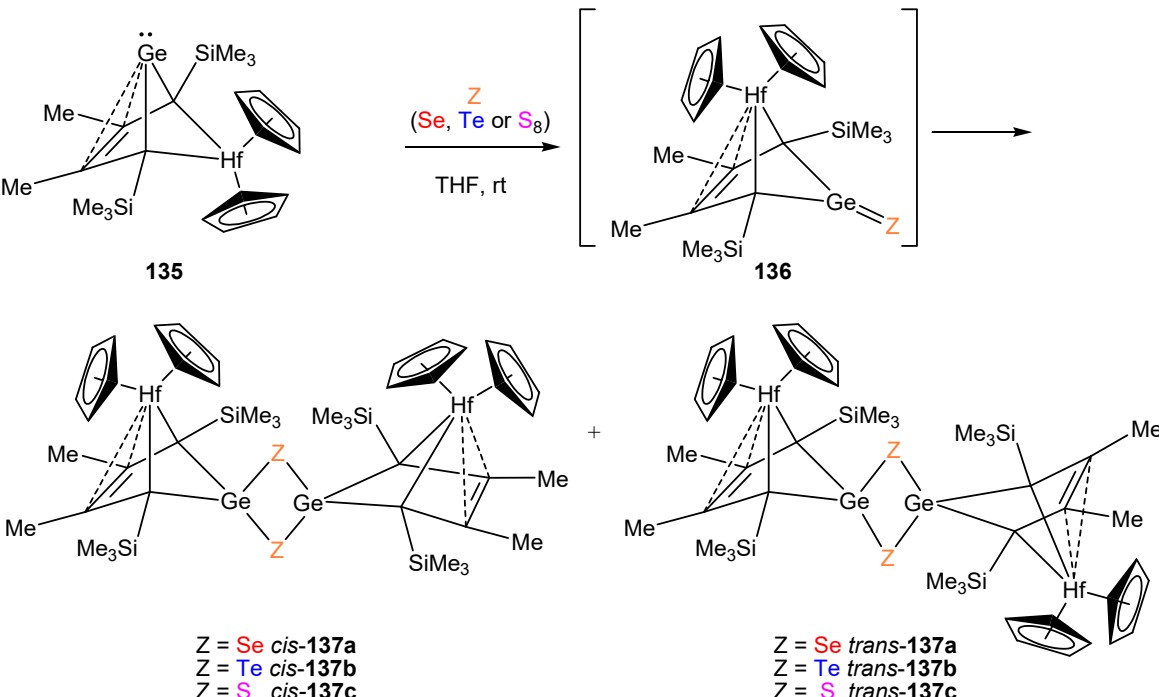

**Scheme 54.** Synthesis of 1,3-diselena-2,4-digermetane **137a** as well as 1,3-ditellura-2,4-digermetane **137b** and 1,3-dithia-2,4-digermetane **137c** by action of elemental Se (Te, S) on hafnocene-based bicyclo [2.1.1]hexene germylene **135**.

**Scheme 55.** Synthesis of 1,3-diselena-2,4-digermetane **137d** by Se–Li exchange in triphenylgermyl lithium **138**.

The treatment of stable cyclic digermenes, 1,2-digermacyclobutene **139** and antiaromatic 1,2-digermacyclobutadiene derivatives **140,** with elemental selenium yielded novel $[Ge_2Se_{2/3}C_x]$ heterocycles, 5,6-diselena-1,4-digermabicyclo [2.1.1]hexane **141**, 5,6,7-triselena-1,4-digermabicyclo [2.2.1]hept-2-ene (1,2,4,3,5-triselenadigermolane) **142** and 5,6-diselena-1,4-digermabicyclo [2.1.1]hex-2-ene **143** (Scheme 56), which should be a convenient procedure for the preparation of the cyclic tetrel selenides [122,123]. These Ge-containing polyselenide products were isolated and characterized using X-ray crystallography.

The formation of selenagermanium heterocycle via the selenation of C=Ge bond was exemplified by the reaction of germabenzene **144a** and 2-germanaphtalene **144b** bearing a Tbt group (Tbt = 2,4,6-tris[bis(trimethylsilyl)methyl]phenyl) with elemental selenium in THF, which resulted in the cyclization and formation of only cyclic triselenides, 1,2,3,4-triselenagermolanes **145a** and **145b** [124] (Scheme 57). These new cyclic triselenides containing a germanium atom were characterized via NMR spectroscopy and elemental analysis.

**Scheme 56.** Synthesis of bicycle Ge-containing polyselenides **141–143** by insertion of elemental Se into Ge=Ge bonds of 1,2-digermacyclobutene **139** and 1,2-digermacyclobutadiene derivatives **140**.

**Scheme 57.** Synthesis of 1,2,3,4-triselenagermolanes **145a,b** by insertion of elemental Se into C=Ge bond.

### 18.3. Formation of Se–P Heterocycles via Insertion of Se into P=P and P–P Bonds

The reactions of P=P systems kinetically stabilized by 2,4,6-tris[bis(trimethylsilyl) methyl]phenyl (Tbt) or 2,6-bis[bis(trimethylsilyl)methyl]- 4-[tris(trimethylsilyl)methyl]phenyl (Bbt) groups—TbtP=PTbt **146a**, TbtP=PFc (Fc = ferrocenyl) **146b** [125] or BbtP=PBbt **146c** [126] —with elemental selenium in the presence of triethylamine, which resulted in the formation of the corresponding selenadiphosphiranes **147** (Scheme 58). The molecular structures of these three-membered heterocyclic compounds were confirmed by spectroscopic analysis.

**Scheme 58.** Synthesis of selenadiphosphiranes **147** by insertion of elemental Se into P=P bond.

The oxidative addition of elemental selenium to the homocyclic pentamer (PhP)$_5$ in refluxing toluene afforded various five-membered P–Se heterocycles. By varying the molar ratio of (PhP)$_5$ to selenium, the different selenaphospholanes **148–150** as well as the red crystalline solid 2,4-diphenyl-1,3,2,4-diselenadiphosphetan-2,4-diselenide, or the so-called Woollins reagent (WR) **151,** were prepared by this method [127–129] (Scheme 59). Karaghiosoff and co-workers extended this oxidative route to other (RP)$_5$ homocyclic pentamers, R=Me, Et, 4-Me$_2$NC$_6$H$_4$ and 4-MeOC$_6$H$_4$ [44,130,131].

**Scheme 59.** Synthesis of selenaphospholanes **148–150** and Woollins reagent **151** by insertion of elemental Se into P–P bond of homocyclic pentamer (PhP)$_5$.

The oxidative addition of elemental selenium to tetraphospholanes (PhP)$_4$CR$_2$ **152a,b** (R = H (a), Me (b)), prepared from (PPh)$_5$ by a reduction with potassium to form the phosphorus chain species, K$_2$P$_4$Ph$_4$, and further by its cyclization with dichloromethane and 2,2-dichloropropane, afforded 4-, 5-, and 6-membered heterocycles—2,3-diselena-1,4-diphospholane **153,** selenadiphosphetanes **154a** and **154b**, tetraselenadiphosphinane **155**, and WR **151** [132] (Scheme 60). The main products were five-membered diselenadiphospholane **153** and four-membered 2-selena-1,3-diphosphetane **154b** prepared from the compound **152** in 94% and 68% yields, correspondingly. Six-membered tetraselenadiphosphinane **155**, WR **151** and four-membered 2-selena-1,3-diphosphetane **154a** were formed only in small quantities. Crystallographic analysis revealed a *trans*-configuration of exocyclic Ph in the formed heterocycles **153** and **154b**.

**Scheme 60.** Synthesis of 4-, 5-, and 6-membered P, Se-containing heterocycles by oxidative addition of elemental Se to tetraphospholanes $(PhP)_4CR_2$ **152a,b**.

### 18.4. Formation of Se–P Heterocycles through Interaction of Se with Methylenephosphorane

The reaction of *tert*-butylarylmethylenetriphenylphosphoranes R-*p*-C$_6$H$_4$C(*t*-Bu)=PPh$_3$ (R = OMe, OPh) **156** with elemental selenium afforded the corresponding five-membered 1,2,4-triselenolanes **157** as *trans*-isomers, four-membered 1,3-diselenetanes **158** and Ph$_3$P=Se [133] (Scheme 61). Triselenolanes **157** were shown to in fact be formed from the selenation of the 1,3-diselenetanes **158**, which were the dimerization products of initially generated selenoketones.

**Scheme 61.** Formation of 1,2,4-triselenolanes **157** and 1,3-diselenetanes **158** by interaction of elemental Se with methylenetriphenylphosphoranes **156**.

## 19. Formation of Heterocycles through Three- and Four-Component Reactions Involving Elemental Selenium

Reactions involving the interaction of elemental selenium with organic substrates can also be found among multicomponent reactions with selenium, since selenium in them directly interacts with an organic intermediate.

2-Aryl-1,3-benzoselenazoles **161** have been formed in a selenium-mediated decarboxylative cyclization of 2-chloronitrobenzenes and chloronitropyridines **159,** and aryl- and hetaryl (pyridine and thiophene) acetic acids **160** under metal-free conditions using

*N*-methylpiperidine (NMP) as a base (Scheme 62). The reactions proceeded in moderate-to-good yields with good functional tolerance [134].

R$^1$ = Me, MeO, CF$_3$, F, Cl, Br; R$^2$ = Me, *t*-Bu, OMe, F, Cl, Br; R$^1$Ar = C$_5$H$_3$N; R$^2$Ar = C$_5$H$_4$N, C$_4$H$_4$S

**Scheme 62.** Synthesis of 2-aryl-1,3-benzoselenazoles **161** by decarboxylative cyclization of 2-chloronitrobenzenes and chloronitropyridines **159**, and aryl- and hetaryl (pyridine and thiophene) acetic acids **160** in the presence of elemental Se.

Another approach to preparation of 2-substituted 1,3-benzoselenazole derivatives **161** consisted in (1) the three-component one-pot reactions of readily available 2-iodoanilines **162**, arylacetic acids **160** or arylmethyl chlorides **163**, as well as selenium powder in the presence of CuBr in DMSO at 120 °C [135], or (2) in the three-component reactions of 2-iodoanilines **162**, aromatic and heteroaromatic aldehydes **164**, as well as elemental Se in DMSO at 120 °C in the presence of a Cu powder catalyst [136]. Substituted 1,3-benzoselenazoles **161** were prepared by these methods in moderate-to-high yields (Scheme 63).

**Scheme 63.** Synthesis of 2-substituted-1,3-benzoselenazoles **161** by three-component reactions of 2-iodoanilines **162**, arylacetic acids **160**, or arylmethyl chlorides **163**, or aldehydes **164** and selenium powder.

The three-component assembly of 1-substituted indoles **165**, aromatic ketones **166** and selenium powder were enabled by the IBr-promoted highly selective double C–H selenylation/annulations. This protocol provided a novel access to a diverse variety of selenopheno [2,3-b]indoles **167** with good efficacy and a broad functional group compatibility [137] (Scheme 64). However, with 2-aryl- and hetaryl-substituted indoles **165,** the same three-component assembly afforded indolyl-substituted benzoselenophenes **56** via the selective formation of one C–C and two C–Se bonds (Scheme 65). Acetophenones with both EWG and EDG were converted to the corresponding products [138]. The reaction mechanism of these two reactions is based on the generation of a 3-vinylindole intermediate and oxidative dual CH selenylation. Annulation in the case of 2-unsubstituted indoles proceeds at the indole substituent, and in the case of 2-arylsubstituted indoles, at the aryl substituent of intermediate 3-vinylindole.

**Scheme 64.** Synthesis of selenopheno [2,3-b]indoles **167** by three-component reaction of 1-substituted indoles **165**, aromatic ketones **166** and elemental Se.

$R^1$ = H, Me, F, Br; $R^2$ = H, Me, Et, Bn; $R^3$ = Ar, thiophen-2-yl
$R^4$ = H, Alk, Hlg, OMe, CN, NO$_2$, OCF$_3$, SO$_2$Me

**Scheme 65.** Synthesis of indolyl-substituted benzoselenophenes **56** by three-component reaction of 2-aryl- and hetaryl-substituted indoles **165**, aromatic ketones **166** and elemental Se.

The base-promoted three-component cascade reaction of *ortho*-functionalized isocyanides **168**, secondary amines **169**, and elemental Se in 1,2-dichloroethane (DCE) at room temperature under metal-free conditions afforded 2-amino-3,1-benzoselenazines **170** in high yields [139] (Scheme 66).

$R^1$ = H, Me, F, Cl, CF$_3$; $R^2$ = OMe, OEt, Ph

**Scheme 66.** Synthesis of 2-amino-3,1-benzoselenazines **170** by three-component cascade reaction of isocyanides **168**, secondary amines **169** and elemental Se.

Alternatively, the three-component mixture of isocyanides **168**, arylamidine hydrochlorides **171** and elemental Se successfully reacted in the presence of *N,N*-diisopropylethylamine (DIPEA) as an efficient base to present a series of 1,2,4-selenadiazol-5-amine derivatives **172** [140] (Scheme 67).

**Scheme 67.** Synthesis of 1,2,4-selenadiazol-5-amine derivatives **172** by three-component reaction of isocyanides **168**, arylamidine hydrochlorides **171** and elemental Se.

Another three-component reaction of isocyanides **168**, alk-2-yn-1-ols **173** and elemental selenium afforded, in the presence of DBU, 2-imino-4-alkylidene-1,3-oxaselenolanes **175** in high yields via the intramolecular addition of selenolate moieties of the generated in situ oxyimidoyl selenoates **174** to the carbon–carbon triple bond (Scheme 68) [141].

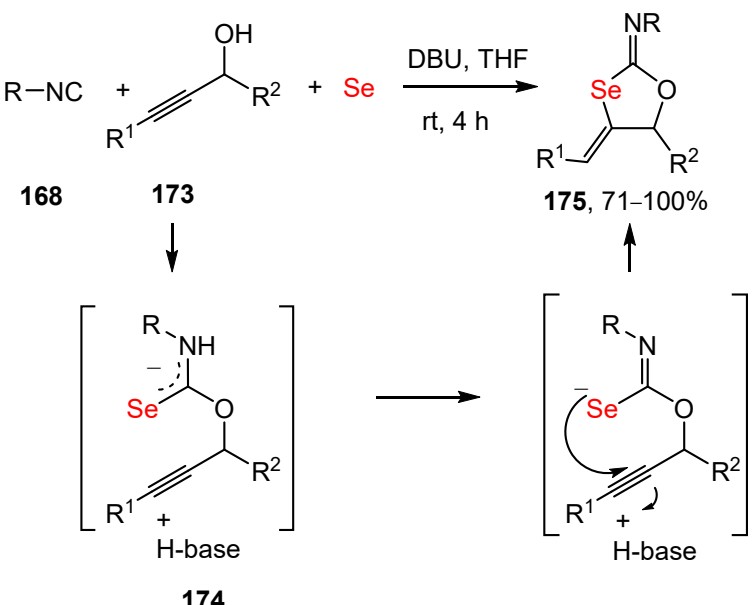

R = 2,6-xylyl, cyclohexyl; $R^1$ = H, Ph, Me, 4-MeC$_6$H$_4$, 4-MeOC$_6$H$_4$, 4-ClC$_6$H$_4$; $R^2$ = H, Ph

**Scheme 68.** Synthesis of 2-imino-4-alkylidene-1,3-oxaselenolanes **175** by three-component reaction of isocyanides **168**, alk-2-yn-1-ols **173** and elemental Se.

2-Substituted naphtho [2,1-d][1,3]selenazoles **178** and naphtho [1,2-d][1,3]selenazoles **179** were prepared in generally high yields for **178** and modest yields for **179** via efficient molecular iodine-catalyzed three-component cascade reactions from naphthalen-2-amine **176** in a case of [1,3]selenazoles **178,** and naphthalen-1-amine **177** in a case of [1,3]selena-zoles **179**, aldehydes **164** and selenium powder [142] (Scheme 69). This approach has the advantages of metal-free conditions, simple operation and available raw materials. The possible reaction mechanism involves the formation of imine intermediates and consequent radical process triggered by iodine radicals.

**Scheme 69.** Syntheses of naphtho [2,1-d][1,3]selenazoles **178** and naphtho [1,2-d][1,3]selenazoles **179** by molecular iodine-catalyzed three-component cascade reactions from naphthalen-2(1)-amines **176** or **177**, aldehydes **164** and elemental Se.

The four-component reaction of (2-benzimidazolyl) acetonitrile **180**, CS$_2$, isothiocyanate **181** and elemental selenium led to a zwitterionic azaselenadithiapentalene **182** [143] (Scheme 70). The structure of the product has been established by XRD analysis. The proposed reaction mechanism comprises the addition of the anion of **180** to CS$_2$ to yield intermediate **A**, which then reacts with selenium to result in intermediate **B**. The ring closure of the latter to **C**, the addition of 4-bromophenyl isothiocyanate to present **D** and cyclization lead to the final product **182**, via tautomerization and protonation. It is worthwhile to mention that the same reaction with elemental sulfur results in tetracyclic [1,3]thiazolo [4′,5′:4,5]pyrimido [1,6-a]benzimidazol-2(3H)-thione (Figure 1) [143].

**Scheme 70.** Formation of zwitterionic azaselenadithiapentalene **182** by four-component reaction of (2-benzimidazolyl) acetonitrile **180**, isothiocyanate **181**, CS$_2$ and elemental Se.

**Figure 1.** [1,3]thiazolo [4′,5′:4,5]pyrimido [1,6-a]benzimidazol-2(3H)-thione.

## 20. Conclusions

In conclusion, metal-to-selenium exchange is the most widely used method for introducing selenium into the heterocycle molecules, with lithium being mainly used as the metal, which is due to the ease of the starting organic compounds lithiation and the ease of further lithium-to-selenium exchange. The synthesis of 1,3-diselenol-2-selones by the interaction of terminal acetylenes with butyllithium, selenium and carbon diselenide can be considered as a variant of this method.

For the construction of five-, six- and seven-membered unsaturated and saturated selenacycles, the exchange of iodine, bromine or chlorine for selenium in a presence of CuI or CuO is also often used. In particular, this method is utilized in the synthesis of ebselen and its analogs from 2-halogenarylamides and selenium.

For the synthesis of a selenium element-containing heterocycles, the introduction or addition of elemental selenium at El–El or C–El bond is mainly used, which distinguishes this methodology from the preparation of C–selenium containing heterocycles, where the introduction of selenium with carbon–carbon bond rupture, or the direct addition of selenium at carbon–carbon bond is impossible.

**Funding:** This research received no external funding.

**Data Availability Statement:** Not applicable.

**Conflicts of Interest:** The authors declare no conflict of interest.

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
