# Peer review of "Elemental Selenium in the Synthesis of Selenaheterocycles"

_inorganics, doi:10.3390/inorganics11070287_

Round 1

Reviewer 1 Report

In the manuscript entitled as “Elemental Selenium in the Synthesis of Selenaheterocycles” the authors describe an overview of the known methods of introduction of elemental selenium into the structures of  unsaturated compounds, to prepare several heteroaromatic heterocycles which contain besides C-Se bonds, Se-N, Se-B, Se-Ge, Se-P bond.

The journal of Inorganics is focuses on synthesis and characterization of inorganic compounds, complexes and materials structure and bonding in inorganic molecular and solid state compounds spectroscopic, among other related topics. In which, this article is a pertinent study, mainly due studies were described heterocycles which contain Se-C, Se-N, Se-B, Se-Ge, Se-P bond and various methods involving metals. Therefore, I recommend this manuscript for publication in Inorganics after consideration of the issues and corrections pointed out below:

·         In the Abstract, correct the word “hehetrocycles”

·         In the Scheme 2, correct the structures “Intermediate”, 9c and 10b.

·         In some places in the text the temperature is “at 200о C” (example, page 3, line 97), correct it. Change to “at 200 °C”.

·         I suggest using the letter Z or ‘’Ch’’ for representation of the chalcogen, and using it in all schemes that reported Se, S or Te. In the Scheme 10, for example it is used M, however in scheme 1, M is Metal.

·         In the Scheme 14, correct the “°C”.

·         Page 15, line 436. Translate the “под действием CO”

·         Page 18, line 528. Translate the (Схема 39)

·         In the Scheme 46. What is the condition of the third arrow?

·         Page 25, line 737. Translate the (Схема 56).

·         In the Scheme 57, change from i, ii and iii to 1), 2) and 3) to standardize whit the other schemes.

·         References, I suggest that you standardize the font and color of the references. Also, I suggest adding https://doi.org/10.1002/adsc.202101227 and https://doi.org/10.1016/j.tet.2022.132752 as references to the article, which were complementary to your review.

Author Response

Response to Reviewer 1 Comments

Point 1: In the Abstract, correct the word “hehetrocycles”

Response 1: In the Abstract the word “hehetrocycles” was changed to “heterocycles”.

Point 2: In the Scheme 2, correct the structures “Intermediate”, 9c and 10b.

Response 2: In the Scheme 2 the structures “Intermediate”, 9c and 10b were corrected.

Point 3: In some places in the text the temperature is “at 200о C” (example, page 3, line 97), correct it. Change to “at 200 °C”.

Response 3: The temperature throughout the text was changed to TTT oC (12 cases).

Point 4: I suggest using the letter Z or ‘’Ch’’ for representation of the chalcogen, and using it in all schemes that reported Se, S or Te. In the Scheme 10, for example it is used M, however in scheme 1, M is Metal.

Response 4: I thank the reviewer for this suggestion. In the schemes 10, 52, 53 I changed representation of the chalcogens S, Se or Te for Z.

Point 5: ·In the Scheme 14, correct the “°C”.

Response 5: Corrected.

Point 6: · Page 15, line 436. Translate the “под действием CO”

Page 18, line 528. Translate the (Схема 39)

Page 25, line 737. Translate the (Схема 56).

Response 6: The omitted translations were inserted.

Point 7: · In the Scheme 46. What is the condition of the third arrow?

Response 7: The conditions of the third arrow (Me3SiBr/CH2Cl2) were inserted in the Scheme 46.

Point 8: · In the Scheme 57, change from i, ii and iii to 1), 2) and 3) to standardize with the other schemes.

Response 8: i, ii and iii were changed to 1), 2) and 3) in the Scheme 57.

Point 9: · References, I suggest that you standardize the font and color of the references.

Response 9: References were arranged according to the Template Inorganics

Point 10: · Also, I suggest adding https://doi.org/10.1002/adsc.202101227 and https://doi.org/10.1016/j.tet.2022.132752 as references to the article, which were complementary to your review.

Response 10: The reviews

  1. Ma, Y.-T.; Liu, M.-Ch.; Zhou, Y.-B.; Wu, H.-Y. Synthesis of organoselenium compounds with elemental selenium. Adv. Synth. Catal. 2021, 363, 5386–5406, doi: 10.1002/adsc.202101227.
  2. Guo, T.; Li, Zh.;  Bi,L.; Fan,L.; Zhang, P. Recent advances in organic synthesis applying elemental selenium. Tetrahedron, 2022, 112, 132752, doi: 10.1016/j.tet.2022.132752

were introduced and discussed in the text.

Reviewer 2 Report

This review by Martynov describes the synthesis of selenaheterocycles using elemental selenium as the chalcogen source. The manuscript is well organized and written. I recommend to accept this work after minor revisions accordingly to the following comments.

1)  In my opinion the references in the introduction are a little bit redundant and contain a series of information not functional to this review (e.g. pharmaceutical applications of selenium compounds are not discussed in the review). The number of references concerning the introduction could be reduced.

2)      The following recent articles may be discussed and properly cited in the review: a) Iodine-Catalyzed Three-Component Cascade Reaction for the Synthesis of Substituted 2-Phenylnaphtho[1,3] selenazoles under Transition-Metal-Free Conditions J. Org. Chem. 2020, 85, 5, 3349–3357; b) Copper-Catalyzed Cascade Multicomponent Reaction of Azides, Alkynes, and Selenium: Synthesis of Ditriazolyl Diselenides, J. Org. Chem. 2023, 88, 7, 4528–4535; c) CuBr2-Catalyzed Annulation of 2-Bromo-N-Arylbenzimidamide with Se/S8 Powder for the Synthesis of Benzo[d]isoselenazole and Benzo[d]isothiazole, J. Org. Chem. 2023, 88, 4, 1963–1976.

Minor corrections:

-          Page 3, line 91: Scheme 3

-          Page 25, line 737: Scheme 56

-          Page 28, line 830: Scheme 62

Author Response

Response to Reviewer 2 Comments

Point 1: In my opinion the references in the introduction are a little bit redundant and contain a series of information not functional to this review (e.g. pharmaceutical applications of selenium compounds are not discussed in the review). The number of references concerning the introduction could be reduced.

Response 1: The Introduction was shortened and number of references was reduced. 

Point 2: The following recent articles may be discussed and properly cited in the review: a) Iodine-Catalyzed Three-Component Cascade Reaction for the Synthesis of Substituted 2-Phenylnaphtho[1,3] selenazoles under Transition-Metal-Free Conditions J. Org. Chem. 2020, 85, 5, 3349–3357; b) Copper-Catalyzed Cascade Multicomponent Reaction of Azides, Alkynes, and Selenium: Synthesis of Ditriazolyl Diselenides, J. Org. Chem. 2023, 88, 7, 4528–4535; c) CuBr2-Catalyzed Annulation of 2-Bromo-N-Arylbenzimidamide with Se/S8 Powder for the Synthesis of Benzo[d]isoselenazole and Benzo[d]isothiazole, J. Org. Chem. 2023, 88, 4, 1963–1976.

Response 2: I have introduced in the text the articles

1.Chen,W.; Zhu,X.; Wang, F.; Yang, Y.; Deng, G.; Liang, Y. Iodine-catalyzed three-component cascade reaction for the synthesis of substituted 2-phenylnaphtho[1,3] selenazoles under transition-metal-free conditions. J. Org. Chem. 2020, 85, 3349–3357, doi: 10.1021/acs.joc.9b03154,

  1. Wang, Q.; Xiao, F.; Huang, Z.; Mao, G.; Deng, G.-J. CuBr2-Catalyzed annulation of 2-bromo-N-arylbenzimidamide with Se/S8 powder for the synthesis of benzo[d]isoselenazole and benzo[d]isothiazole, J. Org. Chem. 2023, 88, 1963–1976, doi: 10.1021/acs.joc.2c02088

and thank the reviewer for these valuable additions.

In the article

Bo-Xun Sun, Xu-Nan Wang, Tai-Gang Fan, Yu-Jian HouYu-Jian Hou  Yun-Tao Shen, and Ya-Min Li. Copper-Catalyzed Cascade Multicomponent Reaction of Azides, Alkynes, and Selenium: Synthesis of Ditriazolyl Diselenides, J. Org. Chem. 2023, 88, 7, 4528–4535, doi: 10.1021/acs.joc.2c03102

synthesis of triazolyl diselenides is presented which is beyond the scope of this review.

Point 3: Page 3, line 91: Scheme 3

Page 25, line 737: Scheme 56

Page 28, line 830: Scheme 62

Response 3: The omitted translations were inserted.

Reviewer 3 Report

The manuscript by Martynov presents an overview of the known methods for introducing elemental selenium into various unsaturated, saturated, and heteroaromatic heterocycles that contain C-Se bonds, Se-N, Se-B, Se-Ge, and Se-P bonds. The work should be of interest of readers of Inorganics. However, I recommend that the authors address some revisions before publication.

1. The last sentence in the abstract should be removed.

2. It is important to note that a review on the same topic has already been published by Lu et al. in 2021 Advanced Synthesis & Catalysis in the Synthesis (10.1002/adsc.202101227). In my opinion, the author did not properly appraise this review by comparing the scope of his work with that of the previously published ones. It would have been beneficial if the author had cited Liu et al.'s review, as it provides valuable insights into previous research in the field.

3. There should be some references added for synthesis of organoselenium compounds. For example, Molecular Catalysis 515 (2021) 111881; Org. Biomol. Chem., 2021,19, 3199-3206; and Chin. J. Org. Chem. 2021, 41, 4798-4807 could be included.

4. In Scheme 2, Compounds 9c and 10b should not be drawn as the same. This error needs to be corrected.

5. In page 3, line 91, the word "Схема" should be spelled correctly as "Scheme". Similarly, similar errors were found on pages 15, lines 431 and 432; page 18, lines 528 and 536; page 25, line 737; and page 28, lines 830 and 834. Therefore, I advise the author to carefully check their manuscript for such typographical errors.

6. The article has too many subsections that make it difficult to understand why the author chose to frame the paper in the way they did. The authors should consider reorganizing the structure of the paper to classify the methods of introducing elemental selenium into organic compounds more reasonably. By doing so, the paper will be easier to read and understand for a broader audience.

Minor editing of English language required

Author Response

Response to Reviewer 3 Comments

Point 1: The last sentence in the abstract should be removed.

Response 1: The last sentence in the abstract is removed.

Point 2: It is important to note that a review on the same topic has already been published by Lu et al. in 2021 Advanced Synthesis & Catalysis in the Synthesis (10.1002/adsc.202101227). In my opinion, the author did not properly appraise this review by comparing the scope of his work with that of the previously published ones. It would have been beneficial if the author had cited Liu et al.'s review, as it provides valuable insights into previous research in the field.

Response 2: The review

Ma, Y.-T.; Liu, M.-Ch.; Zhou, Y.-B.; Wu, H.-Y. Synthesis of Organoselenium Compounds with Elemental Selenium. Adv. Synth. Catal. 2021, 363, 5386–5406, doi: 10.1002/adsc.202101227

was introduced and discussed in the text.

Point 3: There should be some references added for synthesis of organoselenium compounds. For example, Molecular Catalysis 515 (2021) 111881; Org. Biomol. Chem., 2021,19, 3199-3206; and Chin. J. Org. Chem. 2021, 41, 4798-4807 could be included.

Response 3:

These articles

  1. Ruixiang Wang, Huilin Xie, Xiaojing Lai, Jin-Biao Liu, Jinhui Li, Guanyinsheng Qiu. Visible light-enabled iron-catalyzed selenocyclization of N-methoxy-2-alkynylbenzamide. Molecular Catalysis 515 (2021) 111881, doi: 10.1016/j.mcat.2021.111881;
  2. Khin Myat Noe Win, Amol D. Sonawane and Mamoru Koketsu. Synthesis of selenated tetracyclic indoloazulenes via iodine and diorganyl diselenides. Org. Biomol. Chem., 2021, 19, 3199-3206, doi: 10.1039/D1OB00268F;
  3. Yuchao Wang, Jinbiao Liu, Guanyinsheng Qiu, Yu Yang, Hongwei Zhou. Metal-Free Selenizative spiro-Tricyclization of N-Hydroxylethyl-N-arylpropiolamides. Chinese Journal of Organic Chemistry, 2021, 41(12): 4798-4807б, doi:. 10.1039/D1OB00268F)

describe syntheses of heterocycles based on the reactions with diselenides, so they are beyond the scope of this review. Besides, selenium in these heterocycles is present in the substituent (RSe) but not in the structures of the heterocycles.      

Point 4: In Scheme 2, Compounds 9c and 10b should not be drawn as the same. This error needs to be corrected.

Response 4: In the Scheme 2 the structures 9c and 10b as well as intermediates were corrected.

Point 5: In page 3, line 91, the word "Схема" should be spelled correctly as "Scheme". Similarly, similar errors were found on pages 15, lines 431 and 432; page 18, lines 528 and 536; page 25, line 737; and page 28, lines 830 and 834. Therefore, I advise the author to carefully check their manuscript for such typographical errors.

Response 5: All these errors were corrected and the omitted translations were inserted in the text.

Point 6: The article has too many subsections that make it difficult to understand why the author chose to frame the paper in the way they did. The authors should consider reorganizing the structure of the paper to classify the methods of introducing elemental selenium into organic compounds more reasonably. By doing so, the paper will be easier to read and understand for a broader audience.

Response 6: This was an idea of the review – to show the possible ways of using selenium in more detail.

Point 7: Scheme 4 and scheme 5 can be represented in one scheme.

Response 7: Though the parent compounds in the Schemes 4 and 5 are the same these reactions are, in fact, two different processes, one involving Hg-Se exchange, another - Li-Se exchange. And the authors of this work depict these two reactions as two different processes.

Reviewer 4 Report

Manuscript: Inorganics-2448573

Response to authors

The article review intitled “Elemental Selenium in the Synthesis of Selenaheterocycles” is a good topic, however, in my opinión, the text of the article should be revised and restructured with respect to the language english structure and the figures.

The work should be restructured because:

1.- I found an article review intitled “Synthesis of organo-selenium compounds with elemental selenium”  (Adv. Synth. Catal. 2021, 363, 5386-5406) not mentioned in this work, which contains a section called “Synthesis of selenium-containing heterocycles with elemental selenium.” This review should be mentioned and considered as complement in this work.

2.- There are mistakes in several paragraphs of the article review and they should be rewrited.

For instance, the paragraph

An overview of the known methods of introduction of elemental selenium into the structures of various unsaturated, saturated, and heteroaromatic heterocycles which contain besides C-Se bonds, Se-N, Se-B, Se-Ge, Se-P bonds is presented. These hehetrocycles can be accessed 9 through a variety of methods including metal-selenium exchange, exchange of iodine and bromine 10 for selenium, cyclization under action of elemental selenium of 1-(2-bromoaryl)benzimidazoles, 11 polyunsaturated hydrocarbons, acetylenes including propargylic amines, 12 1-chloro-2,2-bis(diethylamino)ethane, 3-halogenaryl amides and aryl amides, diazo-compounds, 13 2-aminoacetophenone, annulation of ethynyl arenes.

Can be changed to, for instance

An overview about the known methods related with the introduction of selenium atoms to form unsaturated, saturated, and heteroaromatic selenacycles containing C-Se, N-Se, B-Se, Ge-Se and P-Se bonds is reported by the use of elemental selenium. These methods include metal, iodine y/or bromine exchange and direct cyclization of 1-(2-bromoaryl)benzimidazoles, polyunsaturated hydrocarbons, acetylenes, propargylic amines, 1-chloro-2,2-bis(diethylamino)ethane, 3-halogenaryl amides, aryl amides, diazo-compounds, 2-aminoacetophenone, annulation of ethynyl arenes.

3.- In several schemes, there are not defined the number of compounds synthetized

For instance, in an scheme, there are     R = H, Me, Et, etc;   R1 = OMe,  Ph, etc

It should be represented as:

134    a       b       c        d,    etc

R  =   H      H     Me      Et,    etc

R1= OMe   Ph   OMe    Ph,   etc

4.- In all the schemes, I recommend to use a colour to represent the selenium atoms to higlight it in the selenacycles.

5.- In scheme 1. The structures 1-3 not represent the magnesium metalacycles

6.- In scheme 2. Compound 10a losse the stereochemistry and compounds 9c and 10b are the same as the intermediate.

7.- Scheme 4 and scheme 5 can be represented in one scheme.

8.- In scheme 9. You should represent the BDSs derived from 31 mentioned in the text

9.- In scheme 10, M is used to represent a metal

10.- The 2,4-digermetane 134d in fig. 54 is diferent to the series 134a-c of scheme 53. The number should be diferent.

Author Response

Response to Reviewer 4 Comments

Point 1: I found an article review intitled “Synthesis of organo-selenium compounds with elemental selenium”  (Adv. Synth. Catal. 2021363, 5386-5406) not mentioned in this work, which contains a section called “Synthesis of selenium-containing heterocycles with elemental selenium.” This review should be mentioned and considered as complement in this work.

Response 1:  The review Ma, Y.-T.; Liu, M.-Ch.; Zhou, Y.-B.; Wu, H.-Y. Synthesis of Organoselenium Compounds with Elemental Selenium. Adv. Synth. Catal. 2021, 363, 5386–5406, doi: 10.1002/adsc.202101227 was introduced and discussed in the text

Point 2: There are mistakes in several paragraphs of the article review and they should be rewrited.

For instance, the paragraph

An overview of the known methods of introduction of elemental selenium into the structures of various unsaturated, saturated, and heteroaromatic heterocycles which contain besides C-Se bonds, Se-N, Se-B, Se-Ge, Se-P bonds is presented. These hehetrocycles can be accessed 9 through a variety of methods including metal-selenium exchange, exchange of iodine and bromine 10 for selenium, cyclization under action of elemental selenium of 1-(2-bromoaryl)benzimidazoles, 11 polyunsaturated hydrocarbons, acetylenes including propargylic amines, 12 1-chloro-2,2-bis(diethylamino)ethane, 3-halogenaryl amides and aryl amides, diazo-compounds, 13 2-aminoacetophenone, annulation of ethynyl arenes.

Can be changed to, for instance

An overview about the known methods related with the introduction of selenium atoms to form unsaturated, saturated, and heteroaromatic selenacycles containing C-Se, N-Se, B-Se, Ge-Se and P-Se bonds is reported by the use of elemental selenium. These methods include metal, iodine y/or bromine exchange and direct cyclization of 1-(2-bromoaryl)benzimidazoles, polyunsaturated hydrocarbons, acetylenes, propargylic amines, 1-chloro-2,2-bis(diethylamino)ethane, 3-halogenaryl amides, aryl amides, diazo-compounds, 2-aminoacetophenone, annulation of ethynyl arenes.

Response 2: Corresponding corrections in translation were introduced in the text.

Point 3: In several schemes, there are not defined the number of compounds synthetized

For instance, in an scheme, there are     R = H, Me, Et, etc;   R1 = OMe,  Ph, etc

It should be represented as:

134    a       b       c        d,    etc

R  =   H      H     Me      Et,    etc

R1= OMe   Ph   OMe    Ph,   etc

Response 3: In my opinion, since the review is devoted to the methods of introducing selenium into the heterocycle, it is quite sufficient to indicate the substituents in the resulting products in the presented form.

Point 4: In all the schemes, I recommend to use a colour to represent the selenium atoms to higlight it in the selenacycles.

Response 4: In all scheme selenium was highlighted with red color.

Point 5: In scheme 1. The structures 1-3 not represent the magnesium metalacycles

Response 5: Scheme 1 was modified to show in one structure both Mg and Al metalacycles.

Point 6: In scheme 2. Compound 10a losse the stereochemistry and compounds 9c and 10b are the same as the intermediate.

Response 6: In the Scheme 2 the structures 9c, 10a, 10b as well as the intermediates were corrected.

Point 7: Scheme 4 and scheme 5 can be represented in one scheme.

Response 7: Though the parent compounds in the Schemes 4 and 5 are the same these reactions are actually two different processes, one one associated with the exchange of Hg-Se, the other – Li-Se. And the authors of this work depict these two reactions as two different processes.

Point 8: In scheme 9. You should represent the BDSs derived from 31 mentioned in the text.

Response 8: Scheme 9 was changed to depict all BDSs 31 mentioned in the text.

Point 9: In scheme 10, M is used to represent a metal

Response 9: In the scheme 10 “M” was changed to “Z” to represent chalcogens.

Point 10: The 2,4-digermetane 134d in fig. 54 is diferent to the series 134a-c of scheme 53. The number should be diferent.

Response 10: The compounds 134a-d have one general heterocyclic structure of 1,3-dichalcogena-2,4-digermetane, that’s why I consider them under one number. Actually, 1,3-ditellura-2,4-digermetane 134b and 1,3-dithia-2,4-digermetane 134c should be given different from 1,3-diselena-2,4-digermetane 134a numbers but it is better to show them as one general structure.  

Round 2

Reviewer 3 Report

Accept in present form

Author Response

I thank the reviewer for accepting the manuscript in present form

Reviewer 4 Report

Please, the paragraph about the recent reference 39 (2021) in addition with reference 53 (2022) should be afther the paragraph about references [40-53].

In the references 39 and 53 (reviews), there are ejamples not considered in this review.

Author Response

The paragraph about the recent reference 39 (2021) in addition with reference 53 (2022) was moved to a place after  the paragraph about references [40-53]. 

Numeration of references was corrected according to this changes.

The changes are highlighted with yellow (the paragraph) and blue (numeration) colors.